# Bacteriophage-Resistant *Salmonella rissen*: An In Vitro Mitigated Inflammatory Response

**DOI:** 10.3390/v13122468

**Published:** 2021-12-09

**Authors:** Rosanna Capparelli, Paola Cuomo, Marina Papaianni, Cristina Pagano, Angela Michela Immacolata Montone, Annarita Ricciardelli, Domenico Iannelli

**Affiliations:** 1Department of Agriculture, University of Naples “Federico II”, 80055 Naples, Italy; paola.cuomo@unina.it (P.C.); marina.papaianni@unina.it (M.P.); domenico.iannelli1935@gmail.com (D.I.); 2Department of Molecular Medicine and Medical Biotechnology, University of Naples “Federico II”, 80055 Naples, Italy; pagano.cris@gmail.com; 3Department of Food Inspection, Istituto Zooprofilattico Sperimentale del Mezzogiorno, 80055 Naples, Italy; angela.montone@izsmportici.it; 4Department of Marine Biotechnology, Stazione Zoologica “Anton Dohrn”, Villa Comunale, 80121 Naples, Italy; annarita.ricciardelli@gmail.com

**Keywords:** phage-resistance, bacterial virulence, lipopolysaccharide

## Abstract

Non-typhoid Salmonella (NTS) represents one of the major causes of foodborne diseases, which are made worse by the increasing emergence of antibiotic resistance. Thus, NTS are a significant and common public health concern. The purpose of this study is to investigate whether selection for phage-resistance alters bacterial phenotype, making this approach suitable for candidate vaccine preparation. We therefore compared two strains of *Salmonella enterica serovar Rissen*: R^R^ (the phage-resistant strain) and R^W^ (the phage-sensitive strain) in order to investigate a potential cost associated with the bacterium virulence. We tested the ability of both R^R^ and R^W^ to infect phagocytic and non-phagocytic cell lines, the activity of virulence factors associated with the main Type-3 secretory system (T3SS), as well as the canonic inflammatory mediators. The mutant R^R^ strain—compared to the wildtype R^W^ strain—induced in the host a weaker innate immune response. We suggest that the mitigated inflammatory response very likely is due to structural modifications of the lipopolysaccharide (LPS). Our results indicate that phage-resistance might be exploited as a means for the development of LPS-based antibacterial vaccines.

## 1. Introduction

*Salmonella enterica* (*S. enterica*) is a Gram-negative bacterium, causing salmonellosis, one of the major threats to human health. Approximately 2500 *Salmonella* serovars have been identified [1] and classified as typhoid or non-typhoid strains, according to host specificity and clinical manifestation [2,3].

Non-typhoid Salmonella (NTS) species—specifically the *S. Typhimurium* or *Enteritidis*—are the most frequent cause of worldwide foodborne gastroenteritis, causing 155,000 deaths every year [4]. In the last decades, *S. Rissen*—so far a rare serotype—has been reported to play a significant role in the onset of foodborne diseases [5]. Even though self-limiting gastroenteritis is the main clinical manifestation of *Salmonella* infection, more severe complications—such as extra-intestinal infections or bacteremia—can occur in immunocompromised patients [5]. Antimicrobial agents are the primary strategy to counteract infectious diseases. However, the increased resistance of *Salmonella* to traditional antimicrobial drugs makes it difficult to prevent Salmonella infections. In this context, vaccination may represent a valid alternative.

Vaccines are designed to prevent infections and reduce the associated morbidity and mortality [6]. In detail, vaccines train the host immune system to recognize and neutralize the pathogen [7], promoting all the steps of the immune response and the production of cellular mediators responsible for the occurrence of the disease symptoms. Further, vaccines initiate a measured immune response, well tolerated by the host, which does not cause immunopathology.

So far, a vaccine protecting against NTS is not yet available [8].

Bacteriophages are viruses specifically targeting bacteria [9]. They represent the most numerous organisms in the biosphere [10], and their competitive coevolution with the host has contributed to the development, by the host, of many resistance mechanisms [11,12]. Bacteria can evade phage attacks by using different strategies. One of these consists of preventing phage adsorption modifying surface structures (usually referred to as phage receptors) [12,13,14]. Such modification has a cost for the bacterium, consisting of altering its virulence. However, a limit for the bacterium may result in an advantage for the host, becoming a potential tool for vaccine development [15,16,17].

The lipopolysaccharide (LPS) plays an important role in both phage adsorption and infection of Gram-negative bacteria [18,19]. In a previous study, we demonstrated differences in the LPS biosynthesis and morphology between the bacteriophage-sensitive (R^W^) and the resistant *S. Rissen* strains (R^R^) [20]. More specifically, we detected reduced expression levels of the *phosphomannomutase1* and *phosphomannomutase2* genes in the R^R^ resistant strain, compared to that of R^W^. Thus, R^R^ was shown to produce a LPS lacking mannose in the O-antigen portion. Furthermore, LPS is a pathogen-associated molecular pattern known to interact with the host Toll-like receptor 4 (TLR4) and activate a strong defense immune response [21]. At the same time, several studies have demonstrated that modified LPSs are poor stimulators of TLR4 and trigger a mild immune response—properties which make them useful for a good candidate vaccine [22]. In this context, we compared in vitro the host inflammatory response following infection with R^R^ or R^W^ strains. R^W^ displayed a stronger inflammation compared to R^R^, potentially attributable to differences in the LPS structure between the two strains. Based on these findings, modified LPS of the phage-resistant *S. Rissen* could represent a potential candidate for vaccine development.

## 2. Materials and Methods

### 2.1. Bacterial Strains and Culture Conditions

Non-typhoid *Salmonella enterica subsp. enterica serovar Rissen* strain R^W^, *Salmonella bongori*, *Salmonella nottingham* and *Salmonella typhimurium* were isolated from a food matrix and characterized by Istituto Zooprofilattico Sperimentale Del Mezzogiorno (Portici, Naples, Italy). The *S. Rissen* strain R^R^ was derived from the R^W^ strain following selection for resistance to phage ϕ1, as previously described [17]. Both the Salmonella strains were grown in Nutrient Broth (Scharlab, S.L., Barcelona, Spain) at 37 °C under vigorous agitation (200 rpm).

### 2.2. Cell Lines and Culture Conditions

AGS (human Caucasian gastric adenocarcinoma) and HT-29 (human Caucasian colon adenocarcinoma) cells were grown in Dulbecco’s modified Eagle’s medium, high glucose (DMEM; Microtech, Pozzuoli, Naples, Italy), and supplemented with 10% fetal bovine serum (FBS; Microtech, Pozzuoli, NA, Italy), 1% penicillin/streptomycin (Gibco, Waltham, MA, USA) and 1% L-glutamine (Gibco, Waltham, MA, USA). U937 (human myeloid leukemia) cells were grown in RPMI-1640 (Microtech, Pozzuoli, NA, Italy) and supplemented with 10% fetal bovine serum (FBS; Microtech, Pozzuoli, NA, Italy), 1% penicillin/streptomycin (Gibco, Waltham, MA, USA) and 1% L-glutamine (Gibco, Waltham, MA, USA). All cell lines were maintained in 5% CO_2_ at 37 °C. U937 cells were induced to differentiate into macrophages by exposing them to phorbol-12-myristate-13-acetate (PMA, 100 ng/mL; Sigma Aldrich, St. Louis, MO, USA) for 48 h. Cells were then washed twice, and the culture medium was replaced with RPMI-1640 without PMA, followed by a resting period of 24 h.

### 2.3. Salmonella Invasion Assay

All the *Salmonella* strains were analyzed for their capacity to colonize the following human cell lines: AGS and HT-29 (non-phagocytic epithelial cell lines), and U937 differentiated into macrophages (phagocytic cell line). Cells were seeded at the density of 1 × 10^6^ per well in 12-well plates and incubated overnight at 37 °C in the presence of 5% CO_2_ and without antibiotics. Salmonella invasion capabilities were evaluated as previously described [20]. Briefly, cell monolayers were infected with 10^8^ CFU/mL in a 12-well plate, at MOI (multiplicity of infection) = 1:100 and incubated for 2 h at 37 °C. After incubation, cell monolayers were washed with PBS (Phosphate buffered saline) and incubated in the presence of 100 μg/mL gentamicin for 30 min. Again, cells were washed with PBS (pH 7.3) and lysed in 1 mL of fresh PBS by scraping. Viable intracellular bacteria were counted after plating serial dilutions in nutrient broth. Results were expressed as the mean ± standard error of the mean (SEM) of the number of intracellular bacteria, expressed in Log_10_ CFU/mL. Experiments were performed in duplicate and repeated at least three times.

### 2.4. Expression Levels of Virulence Genes

The presence of 7 genes related to the virulence of *Salmonella* spp. was detected in R^W^ and R^R^ strains by end-point PCR and electrophoretic run on an automated qiaxcel instrument (Qiagen, Hilden, Germany). Their expression levels were evaluated by qRT-PCR at 2, 4 and 6 hpi on the AGS cell line. The selected virulence factors are related to the presence of prophages (*grvA, gogB, sspH1, sodC1, gtgE*) or plasmids (*spvC*) [21].

### 2.5. Infection on AGS Cell Line

Cells were seeded at the density of 1 × 10^6^ per well in 12-well plates and incubated overnight at 37 °C in the presence of 5% CO_2_, without antibiotics. The next day, cells were infected with R^R^ or R^W^ strains (MOI 1:100) for 2 h, 4 h and 6 h. After infection, cells were washed with PBS, and gentamicin (100 μg/mL) was added for 30 min. AGS cells were then lysed and collected using 1 mL of TRIzol LS reagent (Thermo Fisher Scientific, Waltham, MA, USA); whereas bacteria were collected and lysed using scraping and 500 µL of TRIzol reagent. All the samples were stored at −80 °C until the analysis.

### 2.6. RNA Extraction and RT-qPCR

Total RNA extraction was performed using TRIzol LS reagent (Thermo Fisher Scientific, Waltham, MA, USA) following the manufacturer’s instructions. The quality and quantity of RNA was estimated using NanoDrop 2000 c (Thermo Fisher Scientific, Waltham, MA, USA) and then reverse-transcribed using the high-capacity cDNA Reverse transcription kit (Thermo Fisher Scientific, Waltham, MA, USA). Gene transcript levels were measured using Power SYBR Green PCR Master Mix (Applied Biosystem, Waltham, MA, USA) on a StepOne Real-Time PCR System (Thermo Fisher Scientific, Waltham, MA, USA), according to the standard mode thermal cycling conditions, as indicated by Spatuzza et al. [22]. Relative expression levels of analyzed genes were determined using probes listed in Appendix A. The 2^−ΔΔCT^ method was used to calculate relative changes in gene expression determined from real-time quantitative PCR experiments [23,24]. Target gene expression levels were normalized using housekeeping genes (*recA* for *Salmonella* and *GAPDH* for AGS cell line).

### 2.7. Cytokine Determination by Bio-Plex Assay

Bio-Plex Pro Human Th17 Cytokine Assay (BioRad, Hercules, CA, USA) was performed to detect the level of cytokines in supernatants of AGS cells infected with the R^R^ or R^W^ strain. The assay detects multiple analytes simultaneously in a single sample [25,26].

### 2.8. Western Blotting Analysis

AGS cells were infected with R^W^ or R^R^ strains for 1 h and 2 h. Total proteins were extracted with RIPA lysis buffer (50 mM Tris-HCl, 150 mM NaCl, 0.5% Triton X-100, 0.5% deoxycholic acid, 10 mg/mL leupeptin, 2 mM phenylmethylsulfonyl fluoride and 10 mg/mL aprotinin containing protease and phosphatase inhibitors (Sigma Aldrich, St. Louis, MO, USA)). Samples were quantified using Protein Analysis Dye Reagent Concentrate (BioRad, Hercules, CA, USA). Equal quantities of protein were separated by SDS-PAGE gel and transferred to PVDF membranes using a Trans-Blot Turbo (BioRad). The membranes were blocked with 5% fat-free milk in Tris saline buffer containing 0.1% Tween-20 (TBST) at room temperature for 1 h, incubated with primary antibodies (1:1000) at 4 °C overnight, and incubated with horseradish peroxidase (HRP)-conjugated secondary antibodies (1:2000) (BioRad) at room temperature for 1 h. The signals were detected using the BioRad ChemiDoc MP image sensor after the membranes were soaked in enhanced ECL reagents (ECLTM Prime Western Reagents for Blotting Detection, Amersham, GE Healthcare, Buckinghamshire, UK). Protein bands were detected by chemiluminescence HRP substrate (Millipore, Burlington, MA, USA) and analyzed by Image J software (National Institutes of Health, version 2.1.0/1.53c). Total extracts were normalized using an anti-β-actin antibody. The following antibodies were used for the Western blot analysis: Mouse monoclonal anti-human β-actin antibody and anti-AKT mouse monoclonal antibody were purchased from Santa Cruz Biotechnology (Santa Cruz, CA, USA); anti-pNFKB rabbit monoclonal antibody, anti- NFKB rabbit monoclonal, anti-IKBα rabbit monoclonal antibody, anti-pSTAT3 rabbit monoclonal antibody, anti-STAT3 rabbit monoclonal antibody and anti-pAKT rabbit monoclonal antibody were from Cell Signaling Technology (Danvers, MA, USA). The following were used as secondary antibodies: Goat Anti-Rabbit and Goat Anti-Mouse HRP (BioRad, Hercules, CA, USA). The company and concentrations of all antibodies used are presented in Appendix A.

### 2.9. Statistical Analysis

Statistical analysis was performed using GraphPad Prism 8.0 software (San Diego, CA, USA). All data were compared using two-way ANOVA multiple comparisons. Experimental data are presented as mean ± SD of three independent experiments, performed in triplicate. Statistical analysis was considered statistically significant when *p* < 0.05.

## 3. Results

### 3.1. R^W^ and R^R^ Strains Display the Same Antigenic and Antibiotic Resistance Profiles

The slide agglutination test displayed both strains having the same antigenic determinants of the LPS O-chain (O6, O7) and of flagella (Hf, Hg). The two lines also displayed the same antibiotic profile: both were resistant to cefoxitin and sensitive to the same 20 antibiotics (Appendix A). Recent studies have shown that, in bacteria, acquisition of phage resistance is often associated with loss of antibiotic resistance [27]. The R^R^ strain instead remained resistant to cefoxitin (Appendix A).

### 3.2. The R^W^ and R^R^ Strains Both Exhibit the Same Capacity to Colonize Host Cells

Colonization is a major property of *Salmonella* [28]. Therefore, we tested the two strains (R^R^ and R^W^) for their capacity to colonize the host. The AGS (epithelial gastric adenocarcinoma) and U937 (macrophage) cell lines were incubated for 1 and 2 h with the R^W^ or R^R^ strain. The U937 and AGS cell lines both displayed the capacity to internalize R^W^ and R^R^ bacterial strains to the same extent (Figure 1A). Instead, the HT-29 cell line was not colonized by R^R^ or R^W^, both at 1 and 2 h. In addition, no serovar-specific differences in HT-29 cells’ colonization were observed. We repeated the experiment using additional *Salmonella serovars* (*S. typhimurium, bongori, enteritidis* and *nottingham*). All *Salmonella* strains exhibited no capacity to colonize the HT-29 cell line (Figure 1A,B). According to the literature, studies have shown that, in bacteria, acquisition of phage resistance is associated with defects in the host cell colonization [13]. The R^R^ strain instead behaved exactly as the wildtype (Figure 1A).

### 3.3. R^W^ and R^R^ Strains Exhibit Different Virulence Profiles

The virulence of R^R^ and R^W^ strains was tested, incubating the epithelial AGS cell line with both strains for 2, 4 and 6 h. The expression levels of the virulence genes (*invA, sspH1, sodC1, gtgE, grvA, spvC, gogB*) were measured by RT-qPCR. The *invA* gene—controlling colonization of epithelial cells [29]—was equally expressed in both strains (Figure 2). This result exactly concurs with the one reported in Figure 1A, displaying no difference in the colonization of the epithelial AGS cell line by R^R^ or R^W^. Instead, using the same AGS cell line, significant differences between R^W^ and R^R^ were detected regarding *sodC1, gogB, spvC, sspH1, grvA, gtgE* (Figure 2). *SodC1* protects the bacterium from oxidative burst [30], while gogB protects the host tissue integrity [31]. Both of these genes were expressed at a 100× higher level in R^R^ compared to R^W^. *SpvC* and *sspH1* inhibit NF-kB [32]. *GrvA* and *gtgE* help the bacteria survive in the host [33]. Both of these genes (*spvC-sspH1* and *grvA-gtgE*) were found significantly expressed in R^R^ only at 6 hpi compared to R^W^, while *grvA* was found significantly more expressed in R^R^ already after 4 h of incubation. Taken together, these results indicate: (1) that R^R^ and R^W^ have clear different virulence profiles, and (2) that phage-resistance contributes to bacterial persistence in host cells.

### 3.4. R^W^ and R^R^ Strains Induce a Different Inflammatory Host Response

Bioplex analysis indicated that R^W^ elicits a stronger pro-inflammatory response (higher levels of IL-6, IL-8, G-CSF, MCP-1, MIP-1β and TNF-α) than R^R^. At the same time, both strains produce low and very close levels of IL-10 (Figure 3A). Western blot analysis confirmed these results. NF-kB, Akt and STAT3 were significantly more activated in R^W^ than in R^R^ infected cells (Figure 3B). Interestingly, R^R^ infected cells displayed a reduced expression level of STAT3 and a similar expression level of NF-kB, compared to control cells. NF-kB, Akt and STAT3 pathways are known to play a critical role in the inflammatory response triggered by infections [34,35]. These data show that the phage-resistant strain R^R^ induces a significantly lower pro-inflammatory response than R^W^.

### 3.5. R^W^ and R^R^ Strains Display a Different LpxR and TLR4 Gene Expression Level

The *lpxR* gene is involved in de-acylation of lipid A portion of LPS [36]. A time course RT-PCR experiment displayed that *lpxR* is upregulated in R^R^ and downregulated in R^W^ (Figure 4A). The same experiment displayed also that upregulation of *lpxR* increases together with incubation time (Figure 4A). A high expression of *lpxR* gene in R^R^ could potentially reflect a higher level of de-acylation of the lipid A of the mutant R^R^ strain. Instead, the low level of *lpxR* gene expression of the R^W^ strain suggests that this strain has the classic hexa-acylated lipid A structure. A further confirmation of this conclusion is provided by an independent experiment carried out on additional Salmonella strains (*S. bongori* or *enteritidis*). Again, *lpxR* was found upregulated in R^R^ compared to *S. bongori* and *enteritidis*, which instead expresses a level of *lpxR* comparable to R^W^. The expression of TLR4 is negatively modulated by the presence of deacylated lipid A portion of LPS [37]. In this study, the evidence that *TLR4* gene is downregulated in cells incubated with R^R^ (Figure 4B) represents one more independent proof that R^R^ has acquired resistance to phage ɸ1 by modification of the LPS.

## 4. Discussion

The frequent and often inappropriate use of antibiotics in medicine and intensive farming has favored the selection of antibiotic-resistant bacteria, causing serious consequences for human health. This drawback was further emphasized by the phenomenon of phage-resistant bacteria. Any host fighting against a drug or a parasite inevitably evolves strategies to evade the antagonist and survives.

In the present study, we compare two strains of *Salmonella enterica serovar Rissen*, R^R^ (the phage-resistant strain) and R^W^ (the phage-sensitive strain), in order to know, first, whether the changes associated with the acquisition of phage-resistance affects the host cell physiology and, second, the potential mechanisms responsible for the different host-bacteria interaction.

We firstly evaluated the property of both R^R^ and R^W^ strains to colonize host cells. Both R^R^ and R^W^ were found to colonize AGS and U937 cell lines to the same extent (Figure 1A). To establish the host colonization, Salmonella uses the Type 3 Secretion System (T3SS), a complex machinery encoded by Salmonella pathogenicity islands (SPIs) [37,38], and consists of a cluster of virulence genes [39].

Therefore, to investigate the effect of acquisition of phage-resistance on bacteria virulence, we infected the epithelial AGS cell line separately with one of these two strains and analyzed some of the most representative SPIs-virulence genes. As expected, we observed similar expression levels of *invA*, indicating the same capacity of R^R^ and R^W^ to colonize AGS cells (Figure 2). Instead, marked differences were detected with *gtgE, sodC1* and *grvA*, which were all upregulated in R^R^ compared to R^W^ (Figure 2), suggesting that upregulation might favor the survival of R^R^ in AGS compared to R^W^ [30,40]. Interestingly, we also noted increased expression levels of *gogB, spvC* and *sspH1* genes in the R^R^ strain (Figure 2). These data provide evidence about the capacity of R^R^ to infect the host more efficiently, compared to R^W^, by modulating the host’s innate immune response and surviving longer within the host.

Upon bacterial infection, innate immunity initiates a defensive response, which leads to inflammation. Bacteria have developed strategies to elude the host immune clearance and curb the inflammatory response. Our data indicate that the above statement extends to the R^R^ strain. In accordance with the upregulation of the *sspH1* and *spvC* genes inhibiting NF-KB [32] in the R^R^ strain, we found reduced activation of the nuclear transcription factor-kB (NF-kB) and of its activator Akt in the cells infected with R^R^ (Figure 3B). NF-kB is a critical modulator of inflammation; it initiates the transcription of numerous genes, including cytokines and chemokines [41]. Consistent with this finding, we also detected reduced expression levels of pro-inflammatory cytokines and chemokines in R^R^ infected cells (Figure 3A). More specifically, we observed lower levels of: (1) IL-8, MCP-1 and MIP-1β, responsible for the recruitment of neutrophils, monocytes and lymphocytes at the site of infection [42,43]; (2) IL-6 and TNF-α, directly involved in the early stage of pathogen-induced inflammatory response; and (3) GCS-F, involved in cell growth and differentiation [44,45,46]. Cytokines, in turn, are known to induce the activation of the transcriptional factor STAT3 [47]. Finally, in R^R^ infected cells, we detected reduced activation of the STAT3 protein (Figure 3B). We can conclude that our data indicate the R^R^ strain, as a potential candidate vaccine, modulates the immune response curbing inflammation.

In order to organize the immune defense against the pathogen, evolution has selected ancient receptors that recognize pathogen-associated molecular patterns (PAMPs). Lipopolysaccharide (LPS), the most important Gram-negative PAMP, has also been reported to interact with phage proteins, acting as a phage receptor [48]. Bacteria—including Salmonella species—can alter genes of the LPS biosynthesis pathway, modifying the LPS structure and inhibiting phage adsorption [49]. In a previous work, we demonstrated differences between R^R^ and R^W^ strains in the expression levels of two genes (phosphomannomutase1 and phosphomannomutase2) involved in the LPS biosynthetic pathway. Precisely, a comparative analysis showed that R^R^ produces an LPS lacking mannose sugar in the O-antigen portion [17]. Further, lipid A, a principal component of the LPS [50], induces the inflammatory reaction following interaction with Toll-like receptor 4 (TLR4). Based on these considerations, we investigated whether modifications of the R^R^ phenotype could be attributed to alterations of the LPS-lipid A portion. Salmonella species can synthesize enzymes able to covalently alter the lipid A portion, such as the 3′-O-deacylase, encoded by the *lpxR* gene, which is upregulated in R^R^ (Figure 4A). The 3′-O-deacylated form of the lipid A is a poor stimulator of TLR4 [51], which favors bacteria in evading the host immune response. The downregulation of the TLR4 gene in R^R^ infected cells (Figure 4B) further supports the idea that the phage-resistant strain has acquired the resistance by modifying the LPS structure. Before testing R^R^ in vivo, as a potential candidate vaccine, we will further confirm biochemically that R^R^ displays an altered LPS-lipid A portion.

## 5. Conclusions

In conclusion, this study reports a bacteriophage-resistant *Salmonella rissen* strain, which increases its pathogenicity, most likely due to the potential modification of the LPS-lipid A portion. Literature reports several studies describing vaccines based on modified lipid A portion [52]. Here, we propose a valid alternative to the LPS-synthetic vaccines, consisting of exploiting the capacity of phage-resistant bacteria to modify naturally the LPS-toxic portion.

## Figures and Tables

**Figure 1 viruses-13-02468-f001:**
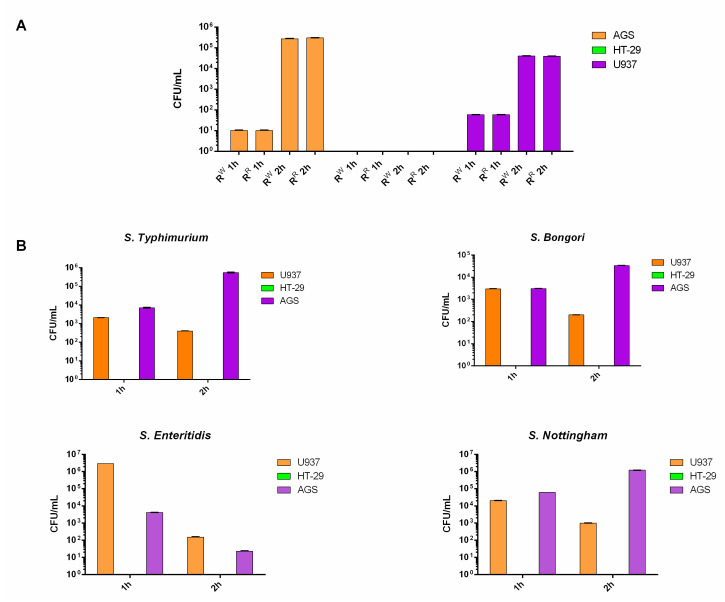
R^R^ and R^W^ host cell colonization. (**A**) AGS, U937 and HT-29 cell lines were infected with the bacteriophage-resistant R^R^ or R^W^ strains for 1 and 2 h. (**B**) AGS, U937 and HT-29 cell lines were infected with *Salmonella typhimurium*, *bongori*, *enteritidis* or *nottingham* for 1 and 2 h. Results are reported as Log_10_ CFU/mL and represent the mean ± SD of three experiments, each performed in triplicate.

**Figure 2 viruses-13-02468-f002:**
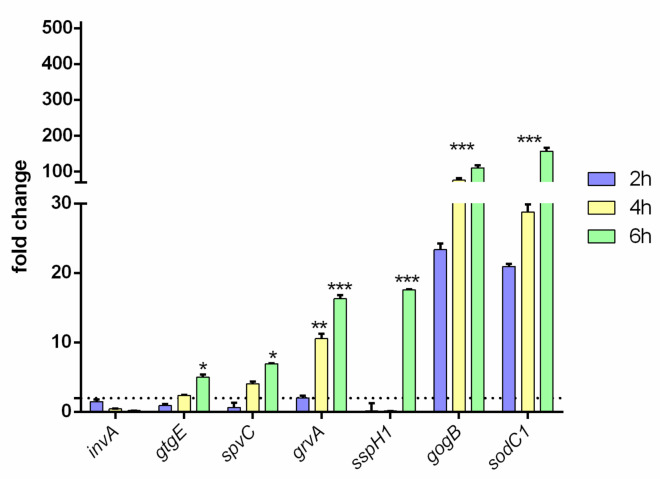
Virulence profile of the R^R^ phage-resistant strain. *invA, gtgE, spvC, grvA, sspH1, gogB* and *sodC1* gene expression levels were measured in the epithelial AGS cell line infected with R^R^ or R^W^ for 2, 4 and 6 h. Results are reported as mean ± SD of three independent experiments, each performed in triplicate and labeled with asterisks (* *p* < 0.05; ** *p* < 0.01; *** *p* < 0.001). Relative gene expression was normalized to R^W^.

**Figure 3 viruses-13-02468-f003:**
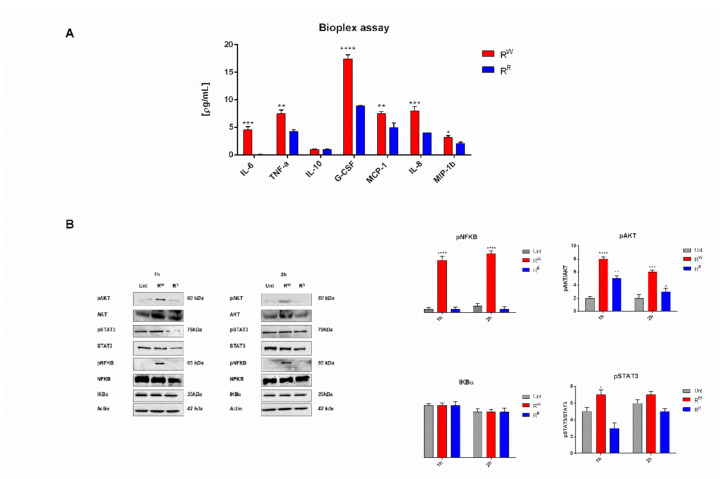
Phage-resistance curbs AGS-induced inflammatory response. (**A**) Cytokines IL-6, TNF-α, IL-10, G-CSF, MCP-1, MIP-1β and IL-8 were measured by Bio-plex assay in AGS cells culture medium after incubation for two hours with R^R^ or R^W^ strains. Results are expressed as pg of cytokines secreted in mL of cell medium. Values were normalized to the basal activity (CTR) and represent mean ± SD of at least three independent experiments, each performed in triplicate (* *p* < 0.05; ** *p* < 0.01; *** *p* < 0.001; **** *p* < 0.0001). (**B**) Western blot and densitometric analysis of the ratio pNF-kB/NF-kB; pAkt/Akt; pSTAT3/STAT3. Actin was used for normalization. Graphs report the result of three independent experiments and represent mean ± SD (* *p* < 0.05; ** *p* < 0.01; *** *p* < 0.001; **** *p* < 0.0001).

**Figure 4 viruses-13-02468-f004:**
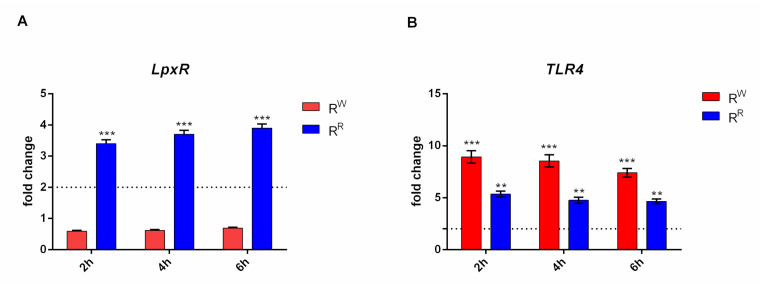
R^R^ and R^W^ strains show a different TLR4 activation. (**A**) Relative gene expression of *lpxR* was determined by quantitative real-time PCR (RT-qPCR), performed on RNA isolated from AGS cells cultured with R^R^ or R^W^ for 2, 4 and 6 h. (*** *p* < 0.001). (**B**) Relative gene expression of *TLR4* was determined by quantitative real-time PCR (RT-qPCR), performed on RNA isolated from AGS cells cultured with R^R^ or R^W^ for 2, 4 and 6 h. All samples were normalized to GAPDH as a reference housekeeping gene. Furthermore, relative gene expression was normalized to basal activity (CTR), in order to obtain relative fold expression. Graphs report the results of at least three independent experiments, represented as means ± SD (** *p* < 0.01; *** *p* < 0.001).

## Data Availability

The data presented in this study era available within the article.

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
