# Peer review of "Bacteriophage-Resistant Salmonella rissen: An In Vitro Mitigated Inflammatory Response"

_viruses, 2021, doi:10.3390/v13122468_

Round 1

Reviewer 1 Report

In the proposed manuscript, Capparelli et al. tested whether changes associated to the acquisition of phage-resistance might be used for vaccine preparation. To this end, phagocytic and non-phagocytic cell lines were infected by a phage sensitive (RS) and phage resistant variant (RR) of the S. enterica serovar Rissen. They conclude that the RR variant has increased pathogenicity due to the potential modification of its LPS_lipid A portion (line 342-343).

It was already demonstrated in the literature that phages resistant variants having decreased in vivo virulence could be used as potential vaccines against S. enterica or S. aureus. (https://doi.org/10.1371/journal.pone.0011720 and https://doi.org/10.1086/648478).

However, it is not clear for the reviewer how a phage resistant variant having increased pathogenicity could be used as a vaccine. Please comment on this point.

Minor comments:

  • The introduction does not cover the use of phage resistant variants as potential vaccines, also different studies already exists on this topic.
  • Line 275: “Generally, phage-resistance is not appreciated as a tool for the production of vaccines because ephemeral, since rapidly neu-tralized by the more efficient bacterial evolution.” A reference would be welcome.
  • Line 286: “Contrary to what expected, selection for phage-resistance did not revert the antibiotic-resistant phenotype (Table S3)”. Why was the resistant variant expected to become less antibiotic resistant? Please clarify.
  • Line 301. “These data provide evidence about the capacity of RR to infect the host more efficiently, compared to RW, by modulating the host innate immune response and surviving longer within the host”. It is not clear how authors went to the conclusion that the resistant variant can survive longer within the host. Please clarify.

Reviewer 2 Report

This study describes the possibility of using phage-resistant Salmonella for vaccine development. It is too early to conclude that phage-resistant strain can be used for vaccine production. Those experiments carried out, including the host cell colonization, virulence profile, pro-inflammatory response, immune-related gene expression, do not sufficient to support vaccine development. Phage-resistant strain should be completely characterized by phenotypic and genotypic properties and further evaluate the alteration in phage-binding receptors and resistance mechanism-based works. Since the interplay between phage-resistant strain and immune response is complex system, more decent experimental approach is needed.

Author Response

We would like to thank the reviewers for their careful reading of the manuscript and their constructive and valuable comments, which provided us the opportunity to modify and improve the text.

Please find below a point-by-point response to all comments.

Response to Reviewer 1

In the proposed manuscript, Capparelli et al. tested whether changes associated to the acquisition of phage-resistance might be used for vaccine preparation. To this end, phagocytic and non-phagocytic cell lines were infected by a phage sensitive (RS) and phage resistant variant (RR) of the S. enterica serovar Rissen. They conclude that the RR variant has increased pathogenicity due to the potential modification of its LPS_lipid A portion (line 342-343).

It was already demonstrated in the literature that phages resistant variants having decreased in vivo virulence could be used as potential vaccines against S. enterica or S. aureus. (https://doi.org/10.1371/journal.pone.0011720 and https://doi.org/10.1086/648478).

However, it is not clear for the reviewer how a phage resistant variant having increased pathogenicity could be used as a vaccine. Please comment on this point.

Answer: We thank the reviewer for his/her comment and for raising this point. As the reviewer suggested, our previous studies (https://doi.org/10.1371/journal.pone.0011720 and https://doi.org/10.1086/648478) demonstrated the capacity of Salmonella enterica or Staphylococcus aureus-phage resistant strains to reduce the virulence in mice, indicating the use of the whole strain as potential candidate vaccine. The present study, instead, is a preliminary in vitro work in which we compared the phage-sensitive S.enterica serovar Rissen (RW) and the phage-resistant S. enterica serovar Rissen (RR) strains, in order to analyze the host cell inflammatory pathway upon bacterial infection and investigate how the acquired phage-resistant phenotype influenced the host physiology. As reported in the manuscript, we observed a stronger inflammation in RW infected cells than in RR ones (Figure 3A and B of the original version of the manuscript), likely due to differences in the LPS structure between the two strains. LPS sensing is essential for pathogen clearance. Gram-negative bacteria can modulate their LPS structure, and specifically the lipid A portion, evading the immune response and promoting bacterial survival and persistence within the host (Needham and Trent, 2013), thus increasing their pathogenicity. As suggested by the upregulation of the LpxR gene (Figure 4A of the original version of the manuscript), RR strain seems to adopt this strategy. LpxR removes 3’-acyl chains of the LPS-lipid A portion, transforming the RR LPS from a full TLR4 agonist to a weak agonist, reducing its ability to induce the inflammatory cell response. In agreement with the reviewer, these features do not make RR a good candidate vaccine as whole organism. However, the increasing interest in vaccines based on the modified LPS-lipid A portion, make the RR strain a potential means for the development of naturally modifies LPS-toxic, as candidate adjuvant vaccine.

Minor comments:

  • The introduction does not cover the use of phage resistant variants as potential vaccines, also different studies already exists on this topic.

Answer: We thank the reviewer for his/her observation. As suggested, we added in the “Introduction” section a general statement about the use of phage resistant strains as potential vaccine. Please, see page 2, lines 61-63.

  • Line 275: “Generally, phage-resistance is not appreciated as a tool for the production of vaccines because ephemeral, since rapidly neu-tralized by the more efficient bacterial evolution.” A reference would be welcome.

Answer: Line 282 was removed. Please, see the revised version of the manuscript.

  • Line 286: “Contrary to what expected, selection for phage-resistance did not revert the antibiotic-resistant phenotype (Table S3)”. Why was the resistant variant expected to become less antibiotic resistant? Please clarify.

Answer: We thank the reviewer for bringing out this point. As reported in the text of the revised version of the manuscript (lines 61-63), the acquisition of the phage-resistance has often a cost for bacteria fitness. Such fitness trade-off might include antibiotic re-sensitization (Alita R. Burmeister et al., 2020; Mangakea and Duerkop; 2020).

  • Line 301. “These data provide evidence about the capacity of RR to infect the host more efficiently, compared to RW, by modulating the host innate immune response and surviving longer within the host”. It is not clear how authors went to the conclusion that the resistant variant can survive longer within the host. Please clarify.

Answer: As indicated in the original version of the manuscript, upon bacterial infection, the phage-resistant strain RR was found to upregulate gtgE gene, responsible for Salmonella survival and replication within macrophages, sodC1 and grvA genes, both cooperating to protect bacteria from the oxidative burst and allow longer survival within the host cell (Rushing and Slauch, 2011; Ho and Slauch, 2001). In addition, it showed higher levels of gogB, spvC and sspH genes, encoding proteins responsible for the inhibition of the transcriptional responses regulating the inflammation (Wang et al., 2020). Taken together, these data suggest the capacity of RR to mitigate the inflammatory response, in order to avoid the host defence mechanism, and survive within the host longer than RW.

References

  1. Burmeister, A.R.; Fortier, A.; Roush, C.; Lessing, A.J.; Bender, R.G.; Barahman, R.; Grant, R.; Chan, B.K.; Turner, P.E. Pleiotropy complicates a trade-off between phage resistance and antibiotic resistance. Proc Natl Acad Sci U S A. 2020;117(21):11207-11216.
  2. Ho, T.D.; Figueroa-Bossi, N.; Wang, M.; Uzzau, S.; Bossi, L.; Slauch, J.M. Identification of GtgE, a novel virulence factor encoded on the Gifsy-2 bacteriophage of Salmonella enterica serovar Typhimurium. J Bacteriol. 2002;184(19):5234-9.
  3. Mangalea, M.R.; Duerkop, B.A. Fitness Trade-Offs Resulting from Bacteriophage Resistance Potentiate Synergistic Antibacterial Strategies. Infect. Immun. 2020, 88, doi:10.1128/IAI.00926-19.
  4. Needham, B.D.; Trent, M.S. Fortifying the barrier: the impact of lipid A remodelling on bacterial pathogenesis. Nat. Rev. Microbiol. 2013, 11:467-481.
  5. Rushing, M.D.; Slauch, J.M. Either periplasmic tethering or protease resistance is sufficient to allow a SodC to protect Salmonella enterica serovar Typhimurium from phagocytic superoxide. Mol Microbiol. 2011;82(4):952-63.
  6. Wang, M.; Qazi, I.H.; Wang, L.; Zhou, G.; Han, H. SalmonellaVirulence and Immune Escape. Microorganisms. 2020; 8(3):407.

Reviewer 2

This study describes the possibility of using phage-resistant Salmonella for vaccine development. It is too early to conclude that phage-resistant strain can be used for vaccine production. Those experiments carried out, including the host cell colonization, virulence profile, pro-inflammatory response, immune-related gene expression, do not sufficient to support vaccine development. Phage-resistant strain should be completely characterized by phenotypic and genotypic properties and further evaluate the alteration in phage-binding receptors and resistance mechanism-based works. Since the interplay between phage-resistant strain and immune response is complex system, more decent experimental approach is needed.

Answer: We thank the reviewer for his/her comments and suggestions.

  1. We agree with the reviewer that the manuscript, in its original version, is not entirely accurate and a bit pretentious. We corrected the text of the revised version of the manuscript according to the reviewer’s observations and minimizing the vaccine topic.
  2. Genotypic and phenotypic properties of the phage resistant strain (RR) have already been characterized and published. Please, see our previous study (doi: 10.1186/s12866-018-1360-z), we have attached to this rebuttal letter.
  3. In vivo studies are the best approach to investigate the host immune response and consequently, in our case, the interaction between the phage-resistant strain (RR) and the immune system. The present study is a preliminary in vitro approach in which we compared the phage-sensitive (RW) and the phage-resistant (RR) enterica serovar Rissen strains, in order to analyze the host cell inflammatory pathway upon bacterial infection and investigate how the acquired phage-resistant phenotype influenced the host cell physiology. However, in agreement with the reviewer’s suggestion, future in vivo studies are necessary to explore the real effect of the RR strain and the plausible use of its modified LPS-lipid A portion as potential candidate adjuvant vaccine. We are just waiting for the authorization by the health minister. Nevertheless, as indicated in lines 347-348, before testing RR in vivo, as a potential candidate vaccine, we will further confirm that RR displays an altered LPS-lipid A portion.

Round 2

Reviewer 1 Report

Authors could significantly improve the overall quality of the manuscript. 

Reviewer 2 Report

This has been properly revised basedon on the comments.